# ACTIVE LEARNING FOR GRAPH NEURAL NETWORKS VIA NODE FEATURE PROPAGATION

## ABSTRACT

Graph Neural Networks (GNNs) for prediction tasks like node classification or edge prediction have received increasing attention in recent machine learning from graphically structured data. However, a large quantity of labeled graphs is difficult to obtain, which significantly limits the true success of GNNs. Although active learning has been widely studied for addressing label-sparse issues with other data types like text, images, etc., how to make it effective over graphs is an open question for research. In this paper, we present an investigation on active learning with GNNs for node classification tasks. Specifically, we propose a new method, which uses node feature propagation followed by K-Medoids clustering of the nodes for instance selection in active learning. With a theoretical bound analysis we justify the design choice of our approach. In our experiments on four benchmark datasets, the proposed method outperforms other representative baseline methods consistently and significantly.

## 1 INTRODUCTION

Graph Neural Networks (GNN) (Kipf & Welling, 2016; Veličković et al., 2017; Hamilton et al., 2017; Wu et al., 2019) have been widely applied in many supervised and semi-supervised learning scenarios such as node classifications, edge predictions and graph classifications over the past few years. Though GNN frameworks are effective at fusing both the feature representations of nodes and the connectivity information, people are longing for enhancing the learning efficiency of such frameworks using limited annotated nodes. This property is in constant need as the budget for labeling is usually far less than the total number of nodes. For example, in biological problems where a graph represents the chemical structure (Gilmer et al., 2017; Jin et al., 2018) of a certain drug assembled through atoms, it is not easy to obtain a detailed analysis of the function for each atom since getting expert labeling advice is very expensive. On the other hand, people can carefully design a small "seeding pool" so that by selecting "representative" nodes or atoms as the training set, a GNN can be trained to get an automatic estimation of the functions for all the remaining unlabeled ones.

Active Learning (AL) (Settles, 2009; Bodó et al., 2011), following this lead, provides solutions that select "informative" examples as the initial training set. While people have proposed various methods for active learning on graphs (Bilgic et al., 2010; Kuwadekar & Neville, 2011; Moore et al., 2011; Rattigan et al., 2007), active learning for GNN has received relatively few attention in this area. Cai et al. (2017) and Gao et al. (2018) are two major works that study active learning for GNN. The two papers both use three kinds of metrics to evaluate the training samples, namely uncertainty, information density, and graph centrality. The first two metrics make use of the GNN representations learnt using both node features and the graph; while they might be reasonable with a good (well-trained) GNN model, the metrics are not informative when the label budget is limited and/or the network weights are under-trained so that the learned representation is not good. On the other hand, graph centrality ignores the node features and might not get the real informative nodes. Further, methods proposed in Cai et al. (2017); Gao et al. (2018) only combine the scores using simple linear weighted-sum, which do not solve these problems principally.

We propose a method specifically designed for GNN that naturally avoids the problems of methods above[1]. Our method select the nodes based on node features propagated through the graph structure,

---

[1] Our code will be released upon acceptance.

making it less sensitive to inaccuracies of representation learnt by under-trained models. Then we cluster the nodes using K-Medoids clustering; K-Medoids is similar to the conventional K-Means, but constrains the centers to be real nodes in the graph. Theoretical results and practical experiments prove the strength of our algorithm.

- We perform a theoretical analysis for our method and study the relation between its classification loss and the geometry of the propagated node features.
- We show the advantage of our method over Coreset (Sener & Savarese, 2017) by comparing the bounds. We also conjecture that similar bounds are not achievable if we use raw unpropagated node features.
- We compare our method with several AL methods and obtain the best performance over all benchmark datasets.

## 2 RELATED WORKS

**Active Learning (AL)** aims at interactively choosing data points from the training pool to maximize model performances, and has been widely studied both in theory (Beygelzimer et al., 2008; Hanneke, 2014) and practice (Settles, 2009; Shen et al., 2017). Recently, Sener & Savarese (2017) proposes to compute a Coreset over the last-layer activation of a convolutional neural network. The method is designed for general-purpose neural networks, and does not take the graph structure into account.

Early works on AL with graph-structured data (Dasarathy et al., 2015; Mac Aodha et al., 2014) study non-parametric classification models with graph regularization. More recent works analyze active sampling under the graph signal processing framework (Ortega et al., 2018; Chen et al., 2016). However, most of these works have focused on the denoising setting where the signal is smooth over the graphs and labels are noisy versions of node features. Similarly, optimal experimental design (Pukelsheim, 2006; Allen-Zhu et al., 2017) can also apply to graph data but primarily deals with linear regression problems, instead of nonlinear classification with discrete labels.

**Graph Neural Networks (GNNs)** (Hamilton et al., 2017; Veličković et al., 2017; Kipf & Welling, 2016) are the emerging frameworks in the recent years when people try to model graph-structured data. Most of the GNN variants follow a multi-layer paradigm. In each layer, the network performs a message passing scheme, so that the feature representation of a node in the next layer could be some neighborhood aggregation from its previous layer. The final feature of a single node thus comprises of the information from a multi-hop neighborhood, and is usually universal and "informative" to be used for multiple tasks. Recent works show the effectiveness of using GNNs in the AL setting. Cai et al. (2017), for instance, proposes to linearly combine uncertainty, graph centrality and information density scores and obtains the optimal performance. Gao et al. (2018) further improves the result by using learnable combination of weights with multi-armed bandit techniques. Instead of combining different metrics, in this paper, we approach the problem by clustering propagated node features. We show that our one-step active design outperforms existing methods based on learnt network represenations, in the small label setting, while not degrading in performance for larger amounts of labeled data.

## 3 PRELIMINARIES

In this section, we describe a formal definition for the problem of graph-based active learning under the node classification setting and introduce a uniform set of notations for the rest of the paper.

We are given a large graph $G = (V, E)$, where each node $v \in V$ is associated with a feature vector $x_v \in \mathcal{X} \subseteq \mathbb{R}^d$, and a label $y_v \in \mathcal{Y} = \{1, 2, ..., C\}$. Let $V = \{1, 2, ..., n\}$, we denote the input features as a matrix $X \in \mathbb{R}^{n \times d}$, where each row represents a node, and the labels as a vector $Y = (y_1, ..., y_n)$. We also consider a loss function $l(\mathcal{M}|G, X, Y)$ that computes the loss over the inputs $(G, X, Y)$ for a model $\mathcal{M}$ that maps $G, X$ to a prediction vector $\hat{Y} \in \mathcal{Y}^n$.

Following previous works on GNN(Cai et al., 2017; Hamilton et al., 2017), we consider the inductive learning setting; i.e., a small part of $Y$ is revealed to the algorithm, and we wish to minimize the loss on the whole graph $l(\mathcal{M}|G, X, Y)$. Specifically, an active learning algorithm $\mathcal{A}$ is initially given the graph $G$ and feature matrix $X$. In step $t$ of operation, it selects a subset $\mathbf{s}^t \subseteq [n] = \{1, 2, ..., n\}$,

and obtains $y_i$ for every $i \in \mathbf{s}^t$. We assume $y_i$ is drawn randomly according to a distribution $\mathbb{P}_{y|x_i}$ supported on $\mathcal{Y}$; we use $\eta_c(v) = \Pr[y = c|v]$ to denote the probability that $y = c$ given node $v$, and $\eta(v) = (\eta_1(v), ..., \eta_C(v))^T$. Then $\mathcal{A}$ uses $G$, $X$ and $y_i$ for $i \in \mathbf{s}^0 \cup \mathbf{s}^1 \cup \cdots \cup \mathbf{s}^t$ as the training set to train a model, using training algorithm $\mathcal{M}$. The trained model is denoted as $\mathcal{M}_{\mathcal{A}_t}$. If $\mathcal{M}$ is the same for all active learning strategies, we can slightly abuse the notation $\mathcal{A}_t = \mathcal{M}_{\mathcal{A}_t}$ to emphasize the focus of active learning algorithms. A general goal of active learning is then to minimize the loss under a given budget $b$:

$$\min_{\mathbf{s}^0 \cup \cdots \cup \mathbf{s}^t} \mathbb{E}[l(\mathcal{A}_t|G, X, Y)] \tag{1}$$

where the randomness is over the random choices of $Y$ and $\mathcal{A}$. We focus on $\mathcal{M}$ being the Graph Neural Networks and their variants elaborated in detail in the following part.

### 3.1 Graph Neural Network Framework

Graph Neural Networks define a multi-layer feature propagation process similar to Multi-Layer Perceptrons (MLPs). Denote the $k$-th layer representation matrix of all nodes as $X^{(k)}$, and $X^{(0)} \in \mathbb{R}^{n \times d}$ are the input node features. Graph Neural Networks (GNNs) differ in their ways of defining the recursive function $f$ for the next-layer representation:

$$X^{(k+1)} \leftarrow f(X^{(k)}; G, \Theta_k), \tag{2}$$

where $\Theta_k$ is the parameter for the $k$-th layer. Naturally, the input $X$ satisfies $X^{(0)} = X$ by definition. **Graph Convolution Network (GCN).** A GCN (Kipf & Welling, 2016) has a specific form of the function $f$ as:

$$X^{(k+1)} \leftarrow \text{ReLU}(SX^{(k)}\Theta_k), \tag{3}$$

where ReLU is the element-wise rectified-linear unit activation function (Nair & Hinton, 2010), $\Theta_k$ is the parameter matrix used for transforming the size of feature representations to a different dimension and $S$ is the normalized adjacency matrix. Specifically, $S$ is defined as:

$$S = (I + D)^{-\frac{1}{2}}(A + I)(I + D)^{-\frac{1}{2}}, \tag{4}$$

where $A$ is the original adjacency matrix associated with graph $G$ and $D$ is the diagonal degree matrix of $A$. Intuitively, this operation updates node embeddings by the aggregation of their neighbors. The added identity matrix $I$ (equivalent to adding self-loops to $G$) acts in a similar spirit to the residual links (He et al., 2016) in MLPs that bypasses shallow-layer representations to deep layers. By applying this operation in a multi-layer fashion, a GCN encourages nodes that are locally related to share similar deep-layer embeddings and prediction results thereafter.

For the classification task, it is normal to stack a linear transformation along with a softmax function to the representation in the final layer, so that each class could have a prediction score. That is,

$$\hat{Y} = \text{softmax}(X^{(K)}\Theta_K), \tag{5}$$

where $\text{softmax}(\boldsymbol{x}) = \exp(\boldsymbol{x})/\sum_{c=1}^{C} \exp(x_c)$ which makes the prediction scores have unit sum of 1 for all classes, and $K$ is the total number of layers. We use the GCN structure as the fixed unified model $\mathcal{M}$ for all the following discussed AL strategies $\mathcal{A}$.

## 4 Active Learning Strategy & Theoretical Analysis

Traditionally, active learning algorithms choose one instance at a time for labeling, i.e., with $|\mathbf{s}^t| = 1$. However, for modern datasets where the numbers of training instances are very large, it would be extremely costly if we re-train the entire system each time when a new label is obtained. Hence we focus on the "batched" one-step active learning setting (Contardo et al., 2017), and select the informative nodes once and for all when the algorithm starts. This is also called the *optimal experimental design* in the literature (Pukelsheim, 2006; Allen-Zhu et al., 2017). Aiming to select the $b$ most representative nodes as the batch, our target (1) becomes:

$$\min_{|\mathbf{s}^0| \leq b} \mathbb{E}[l(\mathcal{A}_0|G, X, Y)]. \tag{6}$$

The node selection algorithm is described in Section 4.1, followed by the loss bound analysis in Section 4.2, and the comparison with a closely related algorithm (K-Center in Coreset (Sener & Savarese, 2017)) in Section 4.3.

## 4.1 Node Selection via Feature Propagation and K-Medoids Clustering

---

**Algorithm 1** Active Learning with Distance-based Clustering

---

**Input:** Node representation matrix $X$, graph structure matrix $G$ and budget $b$
 1: Compute a distance function $d_{X,G}(\cdot, \cdot) : V \times V \to \mathbb{R}$      # for FeatProp: use Eqn. (7)
 2: Perform clustering using $d_{X,G}$ with $b$ centers      # for FeatProp: use K-Medoids
 3: Select $\mathbf{s}$ to be the centers
 4: Obtain labels for $v \in \mathbf{s}$ and train model $\mathcal{M}$
**Output:** Model $\mathcal{M}$

---

We describe a generic active learning framework using distance-based clustering in Algorithm 1. It acts in two major steps: 1) computing a distance matrix or function $d_{X,G}$ using the node feature representations $X$ and the graph structure $G$; 2) applying clustering with $b$ centers over this distance matrix, and from each cluster select the node closest to the center of the cluster. After receiving the labels (given by matrix $Y$) of the selected nodes, we train a graph neural network, specifically GCN, based on $X, G$ and $Y$ for the node classification task. Generally speaking, different options for the two steps above would yield different performance in the down-stream prediction tasks; we detail and justify our choices below and in subsequent sections.

**Distance Function.** Previous methods (Sener & Savarese, 2017; Cai et al., 2017; Gao et al., 2018) commonly use network representations to compute the distance, i.e., $d_{X,G}(v_i, v_j) = \|(X^{(k)})_i - (X^{(k)})_j\|_2$ for some specific $k$. While this can be helpful in a well-trained network, the representations are quite inaccurate in initial stages of training and such distance function might not select the representatitive nodes. Differently, we define the pairwise node distance using the $L_2$ norm of the difference between the corresponding propagated node features:

$$d_{X,G}(v_i, v_j) = \|(S^K X)_i - (S^K X)_j\|_2, \tag{7}$$

where $(M)_i$ denotes the $i$-th row of matrix $M$, and recall that $K$ is the total number of layers. Intuitively, this removes the effect of untrained parameters on the distance, while still taking the graph structure into account.

**Clustering Method.** Two commonly used methods are K-Means (Cai et al., 2017; Gao et al., 2018) and K-Center (Sener & Savarese, 2017)[2]. We propose to apply the K-Medoids clustering. K-Medoids problem is similar to K-Means, but the center it selects must be real sample nodes from the dataset. This is critical for active learning, since we cannot try to label the unreal cluster centers produced by K-Means. Also, we show in Section 4.3 that K-Medoids can obtain a more favorable loss bound than K-Center.

We call our method *FeatProp*, to emphasize the active learning strategy via node feature propagation over the input graph, which is the major difference from other node selection methods.

## 4.2 Theoretical Analysis of Classification Loss Bound

Recall that we use $\|(S^K X)_i - (S^K X)_j\|_2$ to approximate the pairwise distances between the hidden representations of nodes in GCN. Intuitively, representation $S^K X$ resembles the output of a simplified GCN (Wu et al., 2019) by dropping all activation functions and layer-related parameters in the original structure, which introduces a strong inductive bias. In other words, the selected nodes could possibly contribute to the stabilization of model parameters during the training phase of GCN. The following theorem formally shows that using K-Medoids with propagated features can lead to a low classification loss:

**Theorem 1** (informal). *Suppose that the label vector $Y$ is sampled independently from the distribution $y_i \sim \eta(i)$, and the loss function $l$ is bounded by $[-L, L]$. Then under mild assumptions, there exists a constant $c_0$ such that with probability $1 - \delta$ the expected classification loss of $\mathcal{A}_t$ satisfies*

$$\frac{1}{n} l(\mathcal{A}_0 | G, X, Y) \le \frac{c_0}{n} \sum_{i=1}^{n} \min_{j \in \mathbf{s}^0} \|(S^K X)_i - (S^K X)_j\|_2 + \sqrt{\frac{L \log(1/\delta)}{2n}} \tag{8}$$

---

[2]For a group of nodes, K-Center problem aims to find a $\delta$-cover with at most $k$ nodes for smallest possible $\delta$.

To understand Theorem 1, notice that the first term $\sum_{i=1}^{n} \min_{j \in \mathbf{s}^0} \|(S^K X)_i - (S^K X)_j\|_2$ is exactly the target loss of K-Medoids (sum of point-center distances), and the second term $\sqrt{\frac{L \log(1/\delta)}{2n}}$ quickly decays with $n$, where $n$ is the total number of nodes in graph $G$. Therefore the classification loss of $\mathcal{A}_0$ on the entire graph $G$ is mostly dependent on the K-Medoids loss. In practice, we can utilize existing robust initialization algorithms such as Partitioning Around Medoids (PAM) to approximate the optimal solution for K-Medoids clustering.

The assumptions we made in Theorem 1 are pretty standard in the literature, and we illustrate the details in the appendix. While our results share some common characteristics with Sener et al.(Sener & Savarese, 2017), our proof is more involved in the sense that it relates to the translated features $\|(S^K X)_i - (S^K X)_j\|_2$ instead of the raw features $\|(X)_i - (X)_j\|_2$. In fact, we conjecture that using raw feature clustering selection for GCN will not result in a similar bound as in (8): this is because GCN uses the matrix $S$ to diffuse the raw features across all nodes in $V$, and the final predictions of node $i$ will also depend on its neighbors as well as the raw feature $(X)_i$. We could see a clearer comparison in practice in Section 5.2.

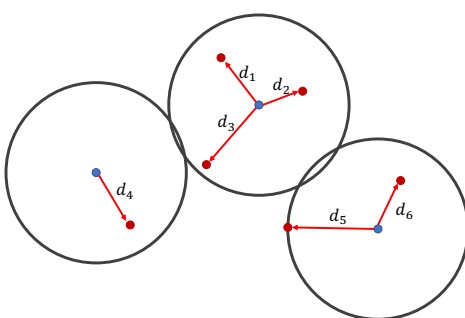

Figure 1: Visualization of Theorem 1. Consider the set of selected points $\mathbf{s}$ and the remaining points in the dataset $[n] \backslash \mathbf{s}$. K-Medoids corresponds to the mean of all red segments in the figure, whereas K-Center corresponds to the max of all red segments in the figure.

### 4.3 WHY NOT K-CENTER

In this subsection we provide justifications on using the K-Medoids clustering method as opposed to Coreset (Sener & Savarese, 2017). The Coreset approach aims to find a $\delta$-cover of the training set. In the context of using propagated features, this means solving

$$\delta = \min_{|\mathbf{s}^0| \leq b} \max_i \min_{j \in \mathbf{s}^0} d_{X,G}(v_i, v_j) = \min_{|\mathbf{s}^0| \leq b} \max_i \min_{j \in \mathbf{s}^0} \|(S^K X)_i - (S^K X)_j\|_2 \qquad (9)$$

We can show a similar theorem as Theorem 1 for the Coreset approach:

**Theorem 2.** *Under the same assumptions as in Theorem 1, with probability $1 - \delta$ the expected classification loss of $\mathcal{A}_t$ satisfies*

$$\frac{1}{n} l(\mathcal{A}_0 | G, X, Y) \leq c_0 \max_i \min_{j \in \mathbf{s}^0} \|(S^K X)_i - (S^K X)_j\|_2 + \sqrt{\frac{L \log(1/\delta)}{2n}} \qquad (10)$$

Let $d_i = \min_{j \in \mathbf{s}^0} \|(S^K X)_i - (S^K X)_j\|_2$. It is easy to see that RHS of Eqn. (8) is smaller than RHS of Eqn. (9), since $\frac{1}{n} \sum_{i=1}^{n} d_i \leq \max_i d_i$. In other words, K-Medoids can obtain a better bound than the K-Center method (see Figure 1 for a graphical illustration). We observe superior performance of K-Medoid clustering over K-Center clustering in our experiments as well (see Section 5.2).

## 5 EXPERIMENT

We evaluate the node classification performance of our selection method on the Cora, Citeseer, and PubMed network datasets (Yang et al., 2016). We further supplement our experiment with an even

denser network dataset CoraFull (Bojchevski & Günnemann, 2017) to illustrate the performance differences of the comparing approaches on a large-scale setting. Table 1 summarizes the dataset statistics.

| Data | # Nodes | # Edges | # Classes | Feature size |
|---|---|---|---|---|
| Cora | 2,708 | 5,429 | 7 | 3,703 |
| Citeseer | 3,327 | 4,732 | 6 | 1,433 |
| PubMed | 19,717 | 44,338 | 3 | 500 |
| CoraFull | 19,793 | 126,842 | 70 | 8,710 |

Table 1: Dataset statistics of different networks.

| | Cora | Citeseer | PubMed | CoraFull |
|---|---|---|---|---|
| FeatProp | 239 | 622 | 1,506 | 13,059 |
| CoresetMIP | 12,260 | 13,257 | OOT | OOT |
| Coreset-greedy | 44 | 46 | 509 | 636 |

Table 2: Comparison of running time of 5 different runs in seconds between our algorithm (FeatProp) and Coreset. OOT denotes out-of-time. Note in order to get a more accurate solution, CoresetMIP costs much more time than Coreset-greedy.

| | Cora | Citeseer | PubMed | CoraFull |
|---|---|---|---|---|
| Random | $59.83 \pm 5.77$ | $48.79 \pm 4.03$ | $71.66 \pm 4.50$ | $10.75 \pm 0.92$ |
| Degree | $63.30 \pm 0.55$ | $35.50 \pm 0.82$ | $60.54 \pm 0.38$ | $10.85 \pm 0.30$ |
| Uncertainty | $48.14 \pm 8.18$ | $39.14 \pm 4.52$ | $64.80 \pm 8.21$ | $6.76 \pm 0.72$ |
| Coreset-greedy | $59.99 \pm 4.59$ | $48.21 \pm 3.78$ | $68.41 \pm 4.50$ | $10.83 \pm 1.28$ |
| CoresetMIP | $55.86 \pm 6.89$ | $46.76 \pm 3.99$ | — | — |
| AGE | $65.01 \pm 2.43$ | $49.65 \pm 5.19$ | $67.96 \pm 2.73$ | $13.52 \pm 0.81$ |
| ANRMAB | $63.71 \pm 4.34$ | $47.29 \pm 3.33$ | $71.06 \pm 4.82$ | $11.40 \pm 0.98$ |
| FeatProp | $\mathbf{74.89 \pm 2.63}$ | $\mathbf{51.03 \pm 2.80}$ | $\mathbf{73.20 \pm 1.81}$ | $\mathbf{14.86 \pm 0.70}$ |

Table 3: Comparison of Macro-F1±standard_deviation averaged over different number of labeled nodes for training. Bold fonts represent the best methods. CorsetMIP does not scale up for PubMed and CoraFull datasets.

We evaluate the Macro-F1 of the methods over the full set of nodes. The sizes of the budgets are fixed for all benchmark datasets. Specifically, we choose to select $10, 20, 40, 80$ and $160$ nodes as the budget sizes. After selecting the nodes, a two-layer GCN [3], with 16 hidden neurons, is trained as the prediction model. We use the Adam (Kingma & Ba, 2014) optimizer with a learning rate of $0.01$ and weight decay of $5 \times 10^{-4}$. All the other hyperparameters are kept as in the default setting ($\beta_1 = 0.9, \beta_2 = 0.999$). To guarantee the convergence of the GCN, the model trained after 200 epochs is used to evaluate the metric on the whole set.

## 5.1 BASELINES

We compared the following methods:

- **Random:** Choosing the nodes uniformly from the whole vertex set.
- **Degree:** Choosing the nodes with the largest degrees. Note that this method does not consider the information of node features.
- **Uncertainty:** Similar to the methods in Joshi et al. (2009), we put the nodes with max-entropy into the pool of instances.

---

[3]In the past semi-supervised setting of citation networks, a two-layer GCN is the optimal structure for the node classification task (Kipf & Welling, 2016).

- **Coreset (Sener & Savarese, 2017):** This method performs a K-Center clustering over the last hidden representations in the network. If time allows (on Cora and Citeseer), a robust mixture integer programming method as in Sener & Savarese (2017) (dubbed CoresetMIP) is adopted. We also apply a time-efficient approximation version (Coreset-greedy) for all of the datasets. The center nodes are then selected into the pool.

- **AGE (Cai et al., 2017):** This method linearly combines three metrics – graph centrality, information density, and uncertainty and select nodes with the highest scores.

- **ANRMAB (Gao et al., 2018):** This method enhances AGE by learning the combination weights of metrics through an exponential multi-arm-bandit updating rule.

- **FeatProp:** This is our method. We perform a K-Medoids clustering to the propogated features (Eqn. (7)), where $X$ is the input node features. In the experiment, we adopts an efficient approximated K-Medoids algorithm which performs K-Means until convergence and select nodes cloesest to centers into the pool.

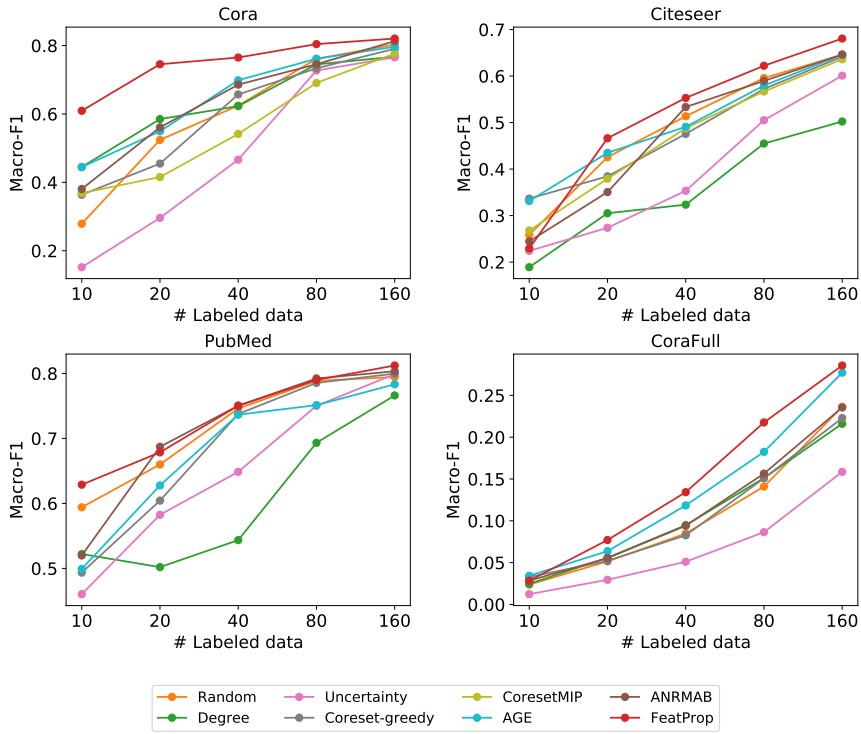

Figure 2: Results of different approaches over benchmark datasets averaged from 5 different runs.

## 5.2 EXPERIMENT RESULTS

In our experiments, we start with a small set of nodes (5 nodes) sampled uniformly at random from the dataset as the initial pool. We run all experiments with 5 different random seeds and report the averaged classification accuracy as the metric. We plot the accuracy vs the number of labeled points. For approaches (Uncertainty, Coreset, AGE and ANRMAB) that require the current status/hidden representations from the classification model, a fully-trained model built from the previous budget pool is returned. For example, if the current budget is 40, the model trained from 20 examples selected by the same AL method is used.

**Main results.** As is shown in Figure 2, our method outperforms all the other baseline methods in most of the compared settings. It is noticeable that AGE and ANRMAB which use uncertainty score as their sub-component can achieve better performances than Uncertainty and are the second best methods in most of the cases. We also show an averaged Macro-F1 with standard deviation across different number of labeled nodes in Table 3. It is interesting to find that our method has the second

smallest standard deviation (Degree is deterministic in terms of node selection and the variance only comes from the training process) among all methods. We conjecture that this is due to the fact that other methods building upon uncertainty may suffer from highly variant model parameters at the beginning phase with very limited labeled nodes.

**Efficiency.** We also compare the time expenses between our method and Coreset, which also involves a clustering sub-routine (K-Center), in Table 2. It is noticeable that in order to make Coreset more stable, CoresetMIP uses an extreme excess of time comparing to Coreset-greedy in the same setting. An interesting fact we could observe in Figure 2 is that CoresetMIP and Coreset-greedy do not have too much performance difference on Citeseer, and Coreset-greedy is even better than CoresetMIP on Cora. This is quite different from the result in image classification tasks with CNNs (Sener & Savarese, 2017). This phenomenon distinguishes the difference between graph node classification with traditional classification problems. We conjecture that this is partially due to the fact that the nodes no longer preserve independent embeddings after the GCN structure, which makes the original analysis of Coreset not applicable.

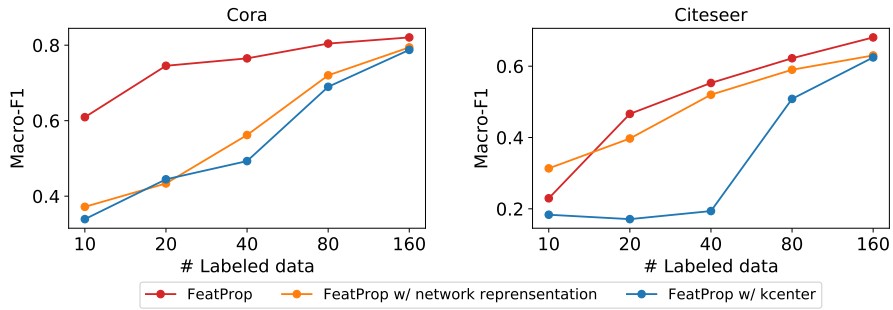

Figure 3: Results of different approaches over benchmark datasets averaged from 5 different runs. Similar to Coreset, the orange line denotes replacing the original distance function in Eqn. (7) with L2 distance from the final GCN layer. The blue line denotes the algorithm replacing the K-Medoids module with K-Center clustering.

**Ablation study.** It is crucial to select the proper distance function and clustering subroutine for FeatProp (Line 1 and Line 2 in Algorithm 1). As is discussed in Section 4.3, we test the differences with the variant of using the L2 distance from the final layer of GCN as the distance function and the one by setting K-Medoids choice with a K-Center replacement. We compare these algorithms in Figure 3. As is demonstrated in the figure, the K-Center version (blue line) has a lower accuracy than the original FeatProp approach. This observation is compatible with our analysis in Section 4.3 as K-Medoids comes with a tighter bound than K-Center in terms of the classification loss. Furthermore, as final layer representations are very sensitive to the small budget case, we observe that the network representation version (orange line) also generally shows a much deteriorated performance at the beginning stage.

Though FeatProp is tailored for GCNs, we could also test the effectiveness of our algorithm over other GNN frameworks. Specifically, we compare the methods over a Simplified Graph Convolution (SGC) (Wu et al., 2019) and obtain similar observations. Due to the space limit, we put the detailed results in the appendix.

# 6 CONCLUSION

We study the active learning problem in the node classification task for Graph Convolution Networks (GCNs). We propose a propagated node feature selection approach (FeatProp) to comply with the specific structure of GCNs and give a theoretical result characterizing the relation between its classification loss and the geometry of the propagated node features. Our empirical experiments also show that FeatProp outperforms the state-of-the-art AL methods consistently on most benchmark datasets. Note that FeatProp only focuses on sampling representative points in a meaningful (graph) representation, while uncertainty-based methods select the active nodes from a different criterion

guided by labels, how to combine that category of methods with FeatProp in a principled way remains an open and yet interesting problem for us to explore.

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

## A    ADDENDUM TO EXPERIMENTS

We also evaluate the methods using the metric of Micro-F1 in Table 4.

|  | Cora | Citeseer | PubMed | CoraFull |
|---|---|---|---|---|
| Random | $65.19 \pm 4.39$ | $57.04 \pm 3.39$ | $73.11 \pm 2.61$ | $21.78 \pm 1.97$ |
| Degree | $68.61 \pm 0.50$ | $46.13 \pm 0.77$ | $68.44 \pm 0.32$ | $25.12 \pm 0.53$ |
| Uncertainty | $58.88 \pm 6.07$ | $46.08 \pm 4.44$ | $68.49 \pm 5.55$ | $14.66 \pm 1.80$ |
| Coreset-greedy | $66.94 \pm 2.87$ | $55.00 \pm 3.09$ | $70.74 \pm 3.15$ | $24.61 \pm 2.35$ |
| CoresetMIP | $63.98 \pm 5.40$ | $55.68 \pm 3.43$ | — | — |
| AGE | $70.07 \pm 1.36$ | $57.27 \pm 4.68$ | $73.80 \pm 1.91$ | $28.35 \pm 1.16$ |
| ANRMAB | $68.62 \pm 3.66$ | $54.90 \pm 3.61$ | $73.98 \pm 3.37$ | $23.14 \pm 1.79$ |
| FeatProp | $\mathbf{77.68 \pm 1.81}$ | $\mathbf{59.36 \pm 1.98}$ | $\mathbf{74.64 \pm 1.49}$ | $\mathbf{28.86 \pm 1.22}$ |

Table 4: Comparison of Micro-F1±standard deviation averaged over different number of labeled nodes for training. Bold fonts represent the best methods. CorsetMIP does not scale up for PubMed and CoraFull datasets.

We evaluate the performances of different active learning methods on a 2-layer SGC (Simplified Graph Convolution) framework. The results can be seen in Figure 4.

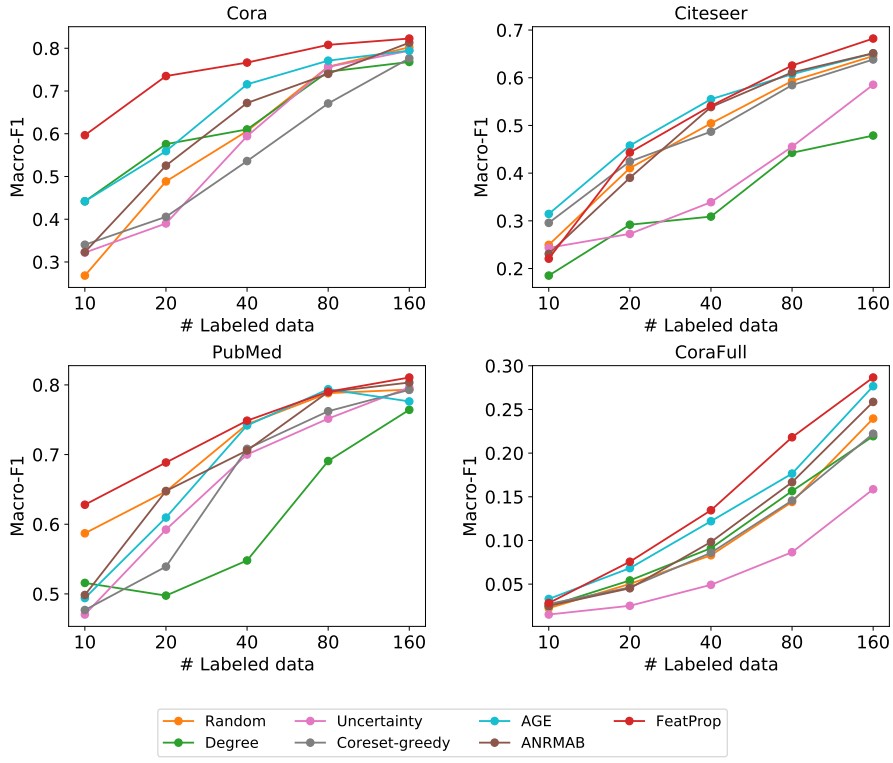

Figure 4: Results of different approaches over benchmark datasets averaged from 5 different runs on an SGC framework.

Figure 5 shows the results of FeatProp on different GNN frameworks. We see that SGC has a slightly inferior performance to GCN since it drops all the activation functions and in-layer parameters, but not too much.

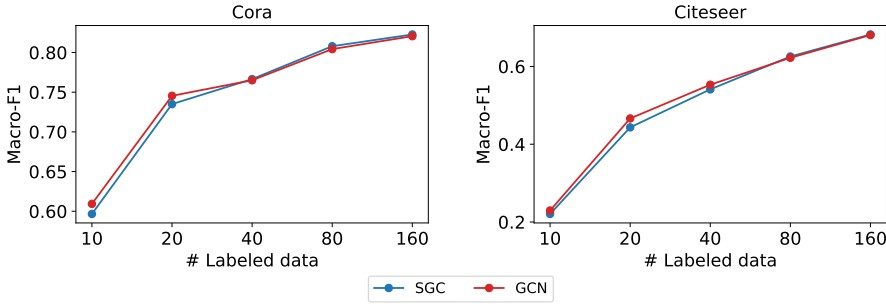

Figure 5: Results of SGC vs GCN over benchmark datasets averaged from 5 different runs by using FeatProp.

# B    PROOF OF THEOREM 1

For simplicity, for any model $\mathcal{M}$ let $(\mathcal{M})_i = (\mathcal{M}(G, X))_i \in \mathbb{R}^C$ be the prediction for node $i$ under input $G, X$, and $(\mathcal{M})_{i,c}$ be the $c$-th element of $(\mathcal{M})_i$ (i.e., the prediction for class $c$). In order to show Theorem 1, we make the following assumptions:

**Assumption 1.** We assume $\mathcal{A}_0$ overfits to the training data. Specifically, we assume the following two conditions: i) $\mathcal{A}_0$ has zero training loss on $\mathbf{s}^0$; ii) for any unlabeled data $(x_i, x_j)$ with $i \notin \mathbf{s}^0$ and $j \in \mathbf{s}^0$, we have $(\mathcal{A}_0)_{i,y_j} \leq (\mathcal{A}_0)_{j,y_j}$ and $(\mathcal{A}_0)_{i,c} \geq (\mathcal{A}_0)_{j,c}$ for all $c \neq y_j$. The second condition states that $\mathcal{A}_0$ achieves a high confidence on trained samples and low confidence on unseen samples. We also assume that the class probabilities are given by a ground truth GCN; i.e., there exists a GCN $\mathcal{M}^*$ that predicts $\Pr[Y_i = c]$ on the entire training set. This is a common assumption in the literature, and (Du et al., 2018) shows that gradient descent provably achieves zero training loss and a precise prediction in polynomial time.

**Assumption 2.** We assume $l$ is Lipschitz with constant $\lambda$ and bounded in $[-L, L]$. The loss function is naturally Lipschitz for many common loss functions such as hinge loss, mean squared error, and cross-entropy if the model output is bounded. This assumption is widely used in DL theory (e.g., Allen-Zhu et al. (2018); Du et al. (2018)).

**Assumption 3.** We assume that there exists a constant $\alpha$ such that the sum of input weights of every neuron is less than $\alpha$. Namely, we assume $\sum_i |(\Theta_K)_{i,j}| \leq \alpha$. This assumption is also present in (Sener & Savarese, 2017). We note that one can make $\sum_i |(\Theta_K)_{i,j}|$ arbitrarily small without changing the network prediction; this is because dividing all input weights by a constant $t$ will also divide the output by a constant $t$.

**Assumption 4.** We assume that ReLU function activates with probability 1/2. This is a common assumption in analyzing the loss surface of neural networks, and is also used in (Choromanska et al., 2015; Kawaguchi, 2016; Xu et al., 2018). This assumption also aligns with observations in practice that usually half of all the ReLU neurons can activate.

With these assumptions in place, we are able to prove Theorem 1.

**Theorem 1 (restated).** Suppose Assumptions 1-4 hold, and the label vector $Y$ is sampled independently from the distribution $y_v \sim \eta(v)$ for every $v \in V$. Then with probability $1 - \delta$ the expected classification loss of $\mathcal{A}_t$ satisfies

$$\frac{1}{n} l(\mathcal{A}_0 | G, X, Y) \leq \frac{(\lambda + L)(\alpha/2)^K}{n} \sum_{i=1}^{n} \min_{j \in \mathbf{s}^0} \|(S^K X)_i - (S^K X)_j\|_2 + \sqrt{\frac{L \log(1/\delta)}{2n}}.$$

*Proof.* Fix $y_j$ for $j \in \mathbf{s}^0$ and therefore the resulting model $\mathcal{A}_0$. Let $i \in V \setminus \mathbf{s}^0$ be any node and $j \in \mathbf{s}^0$. We have

$$
\begin{aligned}
\mathbb{E}_{y \sim \eta(i)}\left[l((\mathcal{A}_0)_i, y)\right] &= \sum_{c=1}^{C} \Pr[Y_i = c] l((\mathcal{A}_0)_{i,c}, c) \\
&= \sum_{c=1}^{C} \Pr[Y_j = c] l((\mathcal{A}_0)_{i,c}, c) + \sum_{c=1}^{C} \left(\Pr[Y_i = c] - \Pr[Y_j = c]\right) l((\mathcal{A}_0)_{i,c}, c).
\end{aligned}
\tag{11}
$$

For the first term we have

$$
\begin{aligned}
\sum_{c=1}^{C} \Pr[Y_j = c] l((\mathcal{A}_0)_{i,c}, c) &= \sum_{c=1}^{C} \Pr[Y_j = c] \left[l((\mathcal{A}_0)_{i,c}, c) - l((\mathcal{A}_0)_{j,c}, c)\right] \\
&\quad + \sum_{c=1}^{C} \Pr[Y_j = c] l((\mathcal{A}_0)_{j,c}, c) \\
&= \sum_{c=1}^{C} \Pr[Y_j = c] \left[l((\mathcal{A}_0)_{i,c}, c) - l((\mathcal{A}_0)_{j,c}, c)\right] \\
&\leq \lambda \sum_{c=1}^{C} \Pr[Y_j = c] \left|(\mathcal{A}_0)_{i,c} - (\mathcal{A}_0)_{j,c}\right|
\end{aligned}
\tag{12}
$$

The last inequality holds from the Lipschitz continuity of $l$. Now from Assumption 1, we have $(\mathcal{A}_0)_{i,c} \geq (\mathcal{A}_0)_{j,c}$ for $c \neq Y_j$ and $(\mathcal{A}_0)_{i,c} \leq (\mathcal{A}_0)_{j,c}$ otherwise. Now taking the expection w.r.t the randomness in ReLU we have

$$
\begin{aligned}
\mathbb{E}_\sigma[(\mathcal{A}_0)_{i,c} - (\mathcal{A}_0)_{j,c}] &= \mathbb{E}_\sigma\left[\sigma((SX^{(K-1)})_i \Theta_K^c) - \sigma((SX^{(K-1)})_j \Theta_K^c)\right] \\
&= \frac{1}{2} \mathbb{E}_\sigma\left[(SX^{(K-1)})_i \Theta_K^c - (SX^{(K-1)})_j \Theta_K^c\right] \\
&\leq \frac{\alpha}{2} \mathbb{E}_\sigma\left[(SX^{(K-1)})_i - (SX^{(K-1)})_j\right] \\
&\leq \cdots \leq \left(\frac{\alpha}{2}\right)^K \left\|(S^K X)_i - (S^K X)_j\right\|.
\end{aligned}
\tag{13}
$$

Here $\mathbb{E}_\sigma$ represents taking the expectation w.r.t ReLU. Now for (12) we have

$$
\begin{aligned}
\mathbb{E}_\sigma\left[\sum_{c=1}^{C} \Pr[Y_j = c] \left|(\mathcal{A}_0)_{i,c} - (\mathcal{A}_0)_{j,c}\right|\right] &= \mathbb{E}_\sigma\left[\sum_{c \neq Y_j} \Pr[Y_j = c] \left((\mathcal{A}_0)_{i,c} - (\mathcal{A}_0)_{j,c}\right)\right] + \\
&\quad \mathbb{E}_\sigma\left[\Pr[Y_j = y_j]\left((\mathcal{A}_0)_{j,y_c} - (\mathcal{A}_0)_{i,c}\right)\right] \\
&\leq \sum_{c=1}^{C} \Pr[Y_j = c] \left(\frac{\alpha}{2}\right)^K \left\|(S^K X)_i - (S^K X)_j\right\| \\
&= \left(\frac{\alpha}{2}\right)^K \left\|(S^K X)_i - (S^K X)_j\right\|.
\end{aligned}
$$

The inequality follows from (13).

Now for the second loss in (11) we use the property that $\mathcal{M}^*$ computes the ground truth:

$$
\left(\Pr[Y_i = c] - \Pr[Y_j = c]\right) l((\mathcal{A}_0)_{i,c}, c) = \left((\mathcal{M}^*)_{i,c} - (\mathcal{M}^*)_{j,c}\right) l((\mathcal{A}_0)_{i,c}, c)
$$

We now use the fact that ReLU activates with probability $1/2$, and compute the expectation:

$$
\begin{aligned}
\mathbb{E}_\sigma\left[\left((\mathcal{M}^*)_{i,c} - (\mathcal{M}^*)_{j,c}\right) l((\mathcal{A}_0)_{i,c}, c)\right] &= \mathbb{E}_\sigma\left[\left((\mathcal{M}^*)_{i,c} - (\mathcal{M}^*)_{j,c}\right)\right] l((\mathcal{A}_0)_{i,c} \\
&= \left(\mathbb{E}_\sigma\left[(\mathcal{M}^*)_{i,c}\right] - \mathbb{E}_\sigma\left[(\mathcal{M}^*)_{j,c}\right]\right) l((\mathcal{A}_0)_{i,c} \\
&= \frac{1}{2^K} \left((S^K X)_i - (S^K X)_j\right) \Theta_1 \Theta_2 \cdots \Theta_K^c l((\mathcal{A}_0)_{i,c} \\
&\leq L \left(\frac{\alpha}{2}\right)^K \left\|(S^K X)_i - (S^K X)_j\right\|.
\end{aligned}
$$

Here $\mathbb{E}_\sigma$ means that we compute the expectation w.r.t randomness in $\sigma$ (ReLU) in $\mathcal{M}^*$. The last inequality follows from definition of $\alpha$, and that $l \in [-L, L]$.

Combining the two parts to (11) and let $j = \arg\min \|(S^K X)_i - (S^K X)_j\|$, we obtain

$$\mathbb{E}_{y \sim \eta(i), \sigma}[l((\mathcal{A}_0)_i, y)] \leq (\lambda + L)(\alpha/2)^K \min_j \|(S^K X)_i - (S^K X)_j\|. \tag{14}$$

Now notice that

$$l(\mathcal{A}_0 | G, X, Y) = \sum_{i \in V \setminus \mathbf{s}^0} l((\mathcal{A}_0)_i, y_i) + \sum_{j \in \mathbf{s}^0} l((\mathcal{A}_0)_j, y_j) = \sum_{i \in V \setminus \mathbf{s}^0} l((\mathcal{A}_0)_i, y_i). \tag{15}$$

Consider the following process: we first get $G, X$ (fixed data) as input, which induces $\eta(i)$ for $i \in [n]$. Note that $\mathcal{M}^*$ gives the ground truth $\eta(i)$ for every $i$ so distributions $\eta(i) \equiv \eta_{X,G}(i)$ are fixed once we obtain $G, X$ [4]. Then the algorithm $\mathcal{A}$ choose the set $\mathbf{s}^0$ to label. After that, we randomly sample $y_j \sim \eta(j)$ for $j \in \mathbf{s}^0$ and use the labels to train model $\mathcal{A}_0$. At last, we randomly sample $y_i \sim \eta(i)$ and obtain loss $l(\mathcal{A}_0 | G, X, Y)$. Note that the sampling of all $y_i$ for $i \in V \setminus \mathbf{s}^0$ is after we fix the model $\mathcal{A}_0$, and knowing exact values of $y_j$ for $j \in \mathbf{s}^0$ does not give any information of $y_i$ (since $\eta(i)$ is only determined by $G, X$). Now we use Hoeffding's inequality (Theorem 3) with $Z_i = l((\mathcal{A}_0)_i, y_i)$; we have $-L \leq Z_i \leq L$ by our assumption, and recall that $|V \setminus \mathbf{s}^0| = n - b$. Let $\delta$ be the RHS of (17), we have that with probability $1 - \delta$,

$$\frac{1}{n-b} \sum_{i \in V \setminus \mathbf{s}^0} l((\mathcal{A}_0)_i, y_i) - \frac{1}{n-b} E_{y \sim \eta(i), \sigma}[l((\mathcal{A}_0)_i, y_i)] \leq \sqrt{\frac{L \log(1/\delta)}{2(n-b)}}.$$

Now plug in (14), multiply both sides by $(n - b)$ and rearrange. We obtain that

$$\sum_{i \in V \setminus \mathbf{s}^0} l((\mathcal{A}_0)_i, y_i) \leq \sum_{i \in V \setminus \mathbf{s}^0} (\lambda + L)(\alpha/2)^K \min_j \|(S^K X)_i - (S^K X)_j\| + \sqrt{\frac{L \log(1/\delta)(n-b)}{2}}. \tag{16}$$

Now note that since the random draws of $y_i$ is completely irrelevant with training of $\mathcal{A}_0$, we can also sample $y_i$ together with $y_j$ for $j \in \mathbf{s}^0$ after receiving $G, X$ and before the training of $\mathcal{A}_0$ ($\mathcal{A}$ does not have access to the labels anyway). So (16) holds for the random drawings of all $y$'s. Now divide both sides of (16) by $n$ and use (15), we have

$$\frac{1}{n} l(\mathcal{A}_0 | G, X, Y) \leq \frac{(\lambda + L)(\alpha/2)^K}{n} \sum_{i=1}^n \min_{j \in \mathbf{s}^0} \|(S^K X)_i - (S^K X)_j\|_2 + \sqrt{\frac{L \log(1/\delta)(n-b)}{2n^2}}$$

$$\leq \frac{(\lambda + L)(\alpha/2)^K}{n} \sum_{i=1}^n \min_{j \in \mathbf{s}^0} \|(S^K X)_i - (S^K X)_j\|_2 + \sqrt{\frac{L \log(1/\delta)}{2n}}.$$

$\square$

## C    PROOF OF THEOREM 2

The same proof as Theorem 1 applies for Theorem 2 using the max of distances instead of averaging. We therefore omit the details here.

## D    HOEFFDING'S INEQUALITY

We attach the Hoeffding's inequality here for the completeness of our paper.

---

[4]To make a rigorous argument, we get the activation of $\mathcal{M}^*$ in this step, meaning that we pass through the randomness of $\sigma$ in $\mathcal{M}^*$.

**Theorem 3** (Hoeffding's Inequality, Hoeffding (1994)). *Suppose $Z_1, ..., Z_n$ are independent random variables such that $a_i \leq Z_i \leq b_i$ almost surely for $1 \leq i \leq n$. Then we have*

$$\Pr\left[\frac{1}{n}\sum_{i=1}^{n} Z_i - E\left[\frac{1}{n}\sum_{i=1}^{n} Z_i\right] > t\right] \leq \exp\left(-\frac{2n^2t^2}{\sum_{i=1}^{n}(b_i - a_i)^2}\right). \tag{17}$$

