# OpenReview forum: "Active Learning Graph Neural Networks via Node Feature Propagation"
_ICLR.cc/2020/Conference — Reject_

### Official Review · AnonReviewer1 · 2019-10-23
**Official Blind Review #1**

**Rating:** 1

**Review:**

The authors propose to incorporate active learning into the graph neural network training and claim some guarantee on the proposed method.

I have some concerns about the correctness of the proof. For theorem 1, how is the Hoeffding applied so that the \sqrt{n} term appears? My worry is naively applying Hoeffding as is done in the proof only gives a bound on a fixed model, but in the theorem A_t is not fixed. You may need to apply a union bound or more sophisticated set cover theory to claim the result. Or if I missed something could the authors add more details on the step using Hoeffding bound to the proof?

Other than that I also feel the assumptions on theorem 1 are way too strong. Especially assumption 2 and 3. They are simply not true in application. The assumptions are so strong that the theorem, even if the proof can be fixed, is not interesting any more.


**Experience Assessment:**

I have read many papers in this area.

**Review Assessment: Checking Correctness Of Derivations And Theory:**

I assessed the sensibility of the derivations and theory.

**Review Assessment: Checking Correctness Of Experiments:**

I assessed the sensibility of the experiments.

**Review Assessment: Thoroughness In Paper Reading:**

I read the paper at least twice and used my best judgement in assessing the paper.

---

> ### Author Response · Authors · 2019-11-11
> **Author Response**
>
> We thank the reviewer for the comments.
>
> About applying Hoeffding’s inequality in our proof: we’ve updated the proof to include more details for this step, and please check out the highlighted part of Appendix B. Briefly, here the randomness in applying Hoeffding’s inequality only comes from the random draws of the hidden labels, $y_i\sim \eta(i)$, for each unlabeled node $i$. This is determined after we fix the graph $G$ and feature matrix $X$. Since our model $A_0$ does not depend on the hidden labels (it cannot see them), making $l(A_0, y_i)$ being independent random variables, we can apply Hoeffding’s inequality without any problem here. Actually, this step of using Hoeffding’s inequality is also present in the coreset paper (Sener & Savarese, 2017, see the last two lines of their proof of Theorem 1).
>
> About the assumptions on theorem 1: our assumptions align with existing works in deep learning/active learning theory which is the general way of real data approximation. For instance, both assumptions 2 and 3 are used in the paper by Sener & Savarese (2017) [3] and Lipschitz (assumption 2) is natural and common for loss functions such as hinge loss, mean squared error, and cross-entropy as long as the model output is bounded. This assumption is also widely used in deep learning theory e.g., [1,2]. As for assumption 3, although the constant $\alpha$ can be unbounded, it can be made arbitrarily small without changing the predicted labels of the network; this is because dividing all input weights by a constant $t$ will also divide the output by a constant $t$. For the other assumptions, assumption 1 assumes a zero training loss, which is a typical setting in neural networks [3]. As for assumption 4, ReLU is activated with probability 1/2, which is justified by the observations in practice that usually half of all the ReLU neurons can activate. As mentioned in the paper, this is a common assumption in the literature. In a related paper on graph learning [4], the authors also assume that the ReLU activations are random.
>
> Moreover, our theoretical analysis only gives worst-case guarantees of our method, and its purpose is to justify our method against other clustering methods (e.g., the coreset approach, clustering the raw features, etc.).
>
> The advantage of our method is evident in our strong experiment results that our simple method can beat previous baselines with elaborately designed heuristics.
>
> We hope that our changes and comments can resolve your question towards our submission - and please reply if you still have further questions, and we would love to provide more details. If we resolve your questions, we are grateful if you can consider updating your review score. Thank you for your time and effort in reviewing our paper!
>
> [1] Allen-Zhu, Z., Li, Y., & Song, Z. (2019). A convergence theory for deep learning via over-parameterization. ICML 2019.
> [2] Du, S. S., & Lee, J. D. (2018). On the power of over-parametrization in neural networks with quadratic activation. ICML 2018.
> [3] Sener, O., & Savarese, S. (2017). Active learning for convolutional neural networks: A core-set approach. arXiv preprint arXiv:1708.00489.
> [4] Xu, K., Li, C., Tian, Y., Sonobe, T., Kawarabayashi, K. I., & Jegelka, S. (2018). Representation Learning on Graphs with Jumping Knowledge Networks. ICML 2018

---

### Official Review · AnonReviewer3 · 2019-10-23
**Official Blind Review #3**

**Rating:** 8

**Review:**

The authors propose an interesting method to actively select samples using the embeddings learned from GNNs. The proposed method combines graph embeddings and clustering to intelligently select new node samples. Theoretical analysis is provided to support the effectiveness and experimental results shows that this method can outperform many other active learning methods.
This paper can be improved on the following aspects:
1.	The proposed method conducts clustering using node embeddings. Although these embeddings have encoded graph structure to some extent, I would suggest explicitly incorporating the graph structure in clustering or at least comparing to a baseline on that. The proposed method conducts embedding learning and clustering in two consecutive but separate steps. It would be interesting to see that the clustering can also leverage the graph information.
2.	It would be better to provide more details about network settings (some hyperparams have already been given in the paper), and more analysis would be helpful. For example, how the number of clusters affects the performance?
3.	Is it possible to create a scenario where there are more labeled data from one cluster but less data from another cluster? In this case, should we still take equal amount of samples from different clusters?


**Experience Assessment:**

I have read many papers in this area.

**Review Assessment: Checking Correctness Of Derivations And Theory:**

I assessed the sensibility of the derivations and theory.

**Review Assessment: Checking Correctness Of Experiments:**

I carefully checked the experiments.

**Review Assessment: Thoroughness In Paper Reading:**

I read the paper thoroughly.

---

> ### Author Response · Authors · 2019-11-11
> **Author Response**
>
> We thank the reviewer for the comments.
>
> We agree that incorporating neighborhood information into the clustering process can also be helpful. For example, we can compute the distance based on a weighted combination of $S, S^2,...$. We will conduct additional experiments on this and report the results in our revised version of the paper.
>
> We train our framework on all datasets with 5 different runs and show the averaged results. We will release our code shortly for people to reproduce our experiments. Currently, we take one sample near each cluster center and so the number of clusters is equal to the label budget. In general, the model performance increases with the number of clusters (labels), as shown in Figure 2.
>
> We agree that varying the number of selected nodes from each cluster is an interesting idea. For example, we may set the number of nodes from each cluster to be proportional to the cluster size, or use hierarchical clustering. It would be meaningful future work to explore more in this aspect.
>
> We hope that our changes and comments can resolve your question towards our submission - and please reply if you still have further questions. Thank you for your time and effort in reviewing our paper!

---

### Official Review · AnonReviewer2 · 2019-11-06
**Official Blind Review #2**

**Rating:** 3

**Review:**

This paper introduces active learning for graphs using graph neural networks

The bound is not very meaningful as it requires unrealistic assumptions and is loose.

Figure 2 shows that even random selection performs quite well compared to this elaborate method.

This Area if research and the data sets don’t seem to have many actual real applications in the world with much impact.

.................................................................\.\\........................................,,..

**Experience Assessment:**

I have published one or two papers in this area.

**Review Assessment: Checking Correctness Of Derivations And Theory:**

I assessed the sensibility of the derivations and theory.

**Review Assessment: Checking Correctness Of Experiments:**

I assessed the sensibility of the experiments.

**Review Assessment: Thoroughness In Paper Reading:**

I made a quick assessment of this paper.

---

> ### Author Response · Authors · 2019-11-11
> **Author Response**
>
> We thank the reviewer for the comments.
>
> [for assumptions]
> We would like to emphasize that our assumptions follow from the common settings in deep learning/active learning theory which is the general way of real data approximation.
>
> For instance, both assumptions 2 and 3 are used in the paper by Sener & Savarese (2017) [3] and Lipschitz (assumption 2) is natural and common for loss functions such as hinge loss, mean squared error, and cross-entropy as long as the model output is bounded. This assumption is also widely used in deep learning theory e.g., [1,2]. As for assumption 3, although the constant $\alpha$ can be unbounded, it can be made arbitrarily small without changing the predicted labels of the network; this is because dividing all input weights by a constant $t$ will also divide the output by a constant $t$. For the other assumptions, assumption 1 assumes a zero training loss, which is a typical setting in neural networks [3]. As for assumption 4, ReLU is activated with probability 1/2, which is justified by the observations in practice that usually half of all the ReLU neurons can activate. As mentioned in the paper, this is a common assumption in the literature. In a related paper on graph learning [4], the authors also assume that the ReLU activations are random.
>
> Moreover, our theoretical analysis only gives worst-case guarantees of our method, and its purpose is to justify our method against other clustering methods (e.g., the coreset approach, clustering the raw features, etc.).
>
> The advantage of our method is evident in our strong experiment results that our simple method can beat previous baselines with elaborately designed heuristics.
>
> Our method is just a clustering of transformed features, which is very easy to implement. It is much simpler than previous active graph learning methods like AGE and ANRMAB, which combine several hand-made heuristics through weighting.
>
> [for Random in Figure 2]
> In Figure 2, please notice that Degree, Uncertainty, Coreset are general active learning methods which cannot leverage  graph-based feature propagationwhile AGE, ANRMAB and our method (FeatProp) are graph-based active learning methods. Our method substantially outperform random sampling  on all the four benchmark datasets in this paper - see Table 4 in Appendix for details.
>
> [for application]
> The main contribution of our paper is to enhance the effectiveness of active learning on graphs. The proposed method is generic and directly applicable to real-world applications where graphical data are available and labeled data are hard to acquire.  For example, our methods can be used to enrich user/item representations in recommendation systems and social networks.
>
> We hope that our changes and comments can resolve your question towards our submission - and please reply if you still have further questions, and we would love to provide more details. If we resolve your questions, we are grateful if you can consider updating your review score. Thank you for your time and effort in reviewing our paper!
>
> [1] Allen-Zhu, Z., Li, Y., & Song, Z. (2019). A convergence theory for deep learning via over-parameterization. ICML 2019.
> [2] Du, S. S., & Lee, J. D. (2018). On the power of over-parametrization in neural networks with quadratic activation. ICML 2018.
> [3] Sener, O., & Savarese, S. (2017). Active learning for convolutional neural networks: A core-set approach. arXiv preprint arXiv:1708.00489.
> [4] Xu, K., Li, C., Tian, Y., Sonobe, T., Kawarabayashi, K. I., & Jegelka, S. (2018). Representation Learning on Graphs with Jumping Knowledge Networks. ICML 2018

---

### Public Comment · ~Le_Wang6 · 2019-10-18
**Need more details about your setting!**

Thanks for your great work, seems that Active Learning works on Graph.
But I don't think it could convince me for the reasons below:
1. Suppose we want 40 nodes, for  K-Medoids you need to choose 40 nodes initially. But in your settings, just start with random 10 nodes?
2. Do you use the original data split setting in the original dataset(Cora, CiteSeer, PubMed)? Why you use 160 as your max training settings for all 4 datasets? It's 140 for Cora,120 for CiteSeer and 60 for PubMed in [1]. It's more convinced that you could reach the same results but fewer nodes in the same setting.
3. In Algorithm I: after step 2 with b centers, how could I get s  as centers, do you mean that s=b?
Hope you could provide mode details about my questions, thanks again!


[1] Kipf T N, Welling M. Semi-supervised classification with graph convolutional networks[J]. arXiv preprint arXiv:1609.02907, 2016.

---

> ### Author Response · Authors · 2019-10-18
> **Clarifications for our paper**
>
> Hi Le,
>
> Thanks for your interest in our paper. For the questions you raised, we would like to make some clarification:
> 1. Sorry for the confusion here. For Uncertainty, Coreset, ANRMAB and AGE, we use an initial “warm-up” set of 5 (instead of 10 - it is a typo in the paper) random nodes for training the model. This is because these methods require a seeding set so that after training on this set, the model could provide node-wise uncertainty and other information which is needed for a later node pool selection. For our one-time selection method, we indeed only need to pick the 40-cluster centers for the selection.
> 2. We do not constrain the methods to the original split. This is the typical case for AL settings where the algorithm is allowed to choose any data points to label instead of relying on a split that already injects some selection bias. And as is known to some readers, the original split is biased [1] and methods may have different results for the average of random splits which is more recommended, we, therefore, use an averaged score (where the split is also not fixed) for evaluation.
> 3. Yes for K-Medoids and Featprop; here the algorithm is requiring that the centers have to be from data points themselves, which leads to $s$ being the set of $b$ centers.
>
> [1] Pitfalls of Graph Neural Network Evaluation https://arxiv.org/abs/1811.05868

---

> > ### Public Comment · ~Le_Wang6 · 2019-10-20
> > **Reply**
> >
> > Thanks for your reply and thanks for your time.
> >
> > Your comment is clear and I could understand your whole algorithm.
> >
> > And for 2: I don't mean that you should use original settings,  you could select nodes from the whole dataset. But maybe the max number of training nodes should be the same, e.g for Cora [14,42,70,98,126,140].

---

> > > ### Author Response · Authors · 2019-10-21
> > > **Reply**
> > >
> > > Thanks for your suggestion! We will consider that in our revised version.

---

### Author Response · Authors · 2019-11-11
**Updates to our paper**

We thank all the reviewers for their time and effort in reviewing our paper. We have revised our paper according to the reviews and updated the version in the OpenReview system.

Updates:
We have revised our assumptions for Theorem 1 and provided more justification towards them.
We include a more detailed proof for the last step of applying Hoeffding’s inequality in proof of Theorem 1.

We hope that these changes can resolve your questions and we are happy to answer any further questions about our paper.

---

### Decision · Program_Chairs · 2019-12-19

**Decision:**

Reject

**Comment:**

The authors propose a method of selecting nodes to label in a graph neural network setting to reduce the loss as efficiently as possible. Building atop Sener & Savarese 2017 the authors propose an alternative distance metric and clustering algorithm. In comparison to the just mentioned work, they show that their upper bound is smaller than the previous art's upper bound. While one cannot conclude from this that their algorithm is better, at least empirically the method appears to have a advantage over state of the art.

However, reviewers were concerned about the assumptions necessary to prove the theorem, despite the modifications made by the authors after the initial round.

The work proposes a simple estimator and shows promising results but reviewers felt improvements like reducing the number of assumptions and potentially a lower bound may greatly strengthen the paper.